# Type III Secretion Effectors with Arginine N-Glycosyltransferase Activity

**DOI:** 10.3390/microorganisms8030357

**Published:** 2020-03-02

**Authors:** Juan Luis Araujo-Garrido, Joaquín Bernal-Bayard, Francisco Ramos-Morales

**Affiliations:** Departamento de Genética, Facultad de Biología, Universidad de Sevilla, 41012 Sevilla, Spain; jaraujo@us.es (J.L.A.-G.); jbbayard@us.es (J.B.-B.)

**Keywords:** glycosyltransferases, type III secretion, effectors, NleB, SseK, *Citrobacter*, Escherichia, Salmonella, death domains

## Abstract

Type III secretion systems are used by many Gram-negative bacterial pathogens to inject proteins, known as effectors, into the cytosol of host cells. These virulence factors interfere with a diverse array of host signal transduction pathways and cellular processes. Many effectors have catalytic activities to promote post-translational modifications of host proteins. This review focuses on a family of effectors with glycosyltransferase activity that catalyze addition of *N*-acetyl-d-glucosamine to specific arginine residues in target proteins, leading to reduced NF-κB pathway activation and impaired host cell death. This family includes NleB from *Citrobacter rodentium*, NleB1 and NleB2 from enteropathogenic and enterohemorrhagic *Escherichia coli*, and SseK1, SseK2, and SseK3 from *Salmonella enterica*. First, we place these effectors in the general framework of the glycosyltransferase superfamily and in the particular context of the role of glycosylation in bacterial pathogenesis. Then, we provide detailed information about currently known members of this family, their role in virulence, and their targets.

## 1. Introduction

Type III secretion systems (T3SSs) are fundamental tools for the interaction between many pathogenic and symbiotic Gram-negative bacteria and their hosts. T3SSs, also known as injectisomes, span the bacterial inner and outer membranes and form injection nanomachines that deliver proteins, known as effectors, into target eukaryotic cells to modulate a variety of cellular functions [1,2]. Effectors have diverse activities and targets, and typically have an N-terminal secretion domain and one or more C-terminal functional domains [3,4]. Effectors can be classified in four non-mutually exclusive categories according to their mechanisms of activity [5]: (i) effectors that mediate their activities through physical interactions with host targets; (ii) effectors that use functional or structural mimicry to alter host cell processes; (iii) effectors that promote post-translational modifications of host proteins; and (iv) effectors with protease activity. Examples of catalytic activities of effectors are ubiquitination, deubiquitination, phosphorylation, dephosphorylation, acetylation, deamidation, ADP ribosylation, cysteine methylation, and glycosylation. This review focuses on a specific family of T3SS effectors with glycosyltransferase activity that are found in *Salmonella enterica* and in attaching/effacing (A/E) pathogens.

*Salmonellae* are facultative intracellular Gram-negative bacterial pathogens belonging to the family Enterobacteriaceae. The genus *Salmonella* includes the species *S. bongori* and *S. enterica*, the latter divided into six subspecies and more than 2500 serovars [6]. *S. enterica* possesses two virulence-related T3SS [7,8,9], T3SS1 and T3SS2, that are encoded by *Salmonella* pathogenicity island 1 (SPI1) and *Salmonella* pathogenicity island 2 (SPI2) (Figure 1), respectively, whereas *S. bongori* lacks SPI2 [10]. The T3SS1 translocates effectors across the host cell plasma membrane that promote *Salmonella* invasion into epithelial cells in the intestine and stimulate localized inflammation [11]. The T3SS2 is expressed intracellularly in response to acidic pH and nutrient limitation found in the lumen of the *Salmonella*-containing vacuole (SCV), a modified phagolysosome which is the typical intracellular niche of this pathogen [12]. T3SS2 effectors are translocated through the SCV membrane, are important in intestinal and disseminated infections, and are required for growth within different host cell types [13]. Together, T3SS1 and T3SS2 secrete more than 40 effectors, some encoded by SPI1 or SPI2, but most encoded outside these islands, in other horizontally acquired regions of the genome [14,15,16,17].

Enterohemorrhagic *Escherichia coli* (EHEC) and enteropathogenic *E. coli* (EPEC) are extracellular human pathogens that attach to the intestinal epithelium and efface brush border microvilli forming A/E lesions [18]. EHEC causes hemorrhagic colitis and hemolytic uremic syndrome, whereas EPEC is a cause of infantile diarrhea. *Citrobacter rodentium* is another A/E pathogen which is the causative agent of murine transmissible colonic hyperplasia [19]. Lesions induced by *C. rodentium* in mice are similar to those caused by EHEC or EPEC in humans and these pathogens share a number of virulence determinants including the locus of enterocyte effacement (LEE) [20,21] (Figure 1). The LEE is a 34 kb pathogenicity island with 41 genes that encode all the structural components of a T3SS, seven of its effectors and their chaperones, proteins involved in adherence, and regulatory proteins [22]. Many effectors are encoded outside the LEE, and the total number of effectors is at least 29 in *C. rodentium*, 22 in EPEC, and 39 in EHEC [22]. A second gene cluster, potentially encoding a T3SS and known as ETT2, for *E. coli* T3SS 2, is present in many *E. coli* strains [23,24]. ETT2 resembles the *Salmonella* T3SS1 but in most cases it has been subjected to mutational attrition and there is no evidence for the secretion of effectors by this system [25]. However, this region is intact in some strains of *E. coli* and *Escherichia albertii*, and appears to be important for bacterial pathogenesis [25,26,27].

## 2. Glycosyltransferases

Glycosyltransferases (GTs) catalyze the formation of glycoside bonds using an activated donor sugar substrate that contains a nucleoside phosphate or a lipid phosphate leaving group [29]. Enzymes that use sugar mono- or diphosphonucleotides as donors are known as Leloir GTs [30]. Non-Leloir GTs use non-nucleotide donors, such as polyprenol pyrophosphates, polyprenol phosphates, sugar-1-phosphates, or sugar-1-pyrophosphates. Sugars can be transferred to a variety of biomolecules including glycans, lipids, proteins, peptides, and small molecules. The resulting glycosidic linkage can be an O-, S-, N- or C- covalent bond connecting a monosaccharide to another residue.

### 2.1. Classification

Traditionally, GTs were classified on the basis of their donor, acceptor, and product specificity, according to the recommendations of the International Union of Biochemistry and Molecular Biology (IUBMB), under the Enzyme Commission number EC.2.4. Given the limitations of this system, a new scheme that involves the classification of GTs into families on the basis of amino acid sequence similarities is now accepted [31,32]. Today, the carbohydrate active enzymes (CAZy) database (http://www.cazy.org) includes over 620,000 GT modules classified in more than 100 families and more than 13,000 nonclassified modules.

Depending on their mechanism of transfer, GTs can be classified as inverting or retaining, resulting in inversion (α- > β) or retention (α- > α) of the stereochemistry of the donor’s anomeric bond during the transfer, respectively [33]. Theoretical and experimental studies support a single-displacement S_N_2 (substitution, nucleophilic, bimolecular)-like mechanism for inverting GTs [29]. This mechanism involves the formation of an oxocarbenium ion transition state (Figure 2). In contrast, the reaction mechanism of retaining glycosylation is controversial [34]. A double-displacement mechanism, in which the sugar is first transferred to the enzyme and then transferred to the acceptor, was suggested by analogy to retaining glycosidases [35]. Some experimental support was provided for this mechanism [36,37], particularly for the GT6 family members that have a carboxylate properly oriented to act as a catalytic nucleophile. But this is not the case for most retaining GTs, leading to the proposal of an alternative internal return S_N_i (substitution, nucleophilic, internal)-like mechanism, also termed front-face mechanism, in which the nucleophilic hydroxy group of the acceptor attacks the anomeric carbon atom from the same side from which the leaving group departs [38,39]. An orthogonal mechanism has also been proposed that differs from the S_N_i mechanism in the reaction profile and the time of bond formation and breakage [40] (Figure 2).

### 2.2. Structural Studies

GT structures (Figure 3) revealed that the majority of these enzymes fall into two general folds, called GT-A and GT-B, each of which includes members that are retaining and inverting GTs. This led to a classification of GTs in four clans: clan I for inverting enzymes with the GT-A fold, clan II for inverting enzymes with the GT-B fold, clan III for retaining enzymes with the GT-A fold, and clan IV for retaining enzymes with the GT-B fold [32]. Both fold types contain a Rossmann fold [41], a nucleotide-binding domain responsible for donor nucleotide recognition. A second domain is usually responsible for acceptor recognition and is more variable. Enzymes with the GT-B type fold contain two Rossmann-like folds separated by a deep wide crevice [42]. However, not all GTs fit into these clans. A third fold type, called GT-C fold, was predicted using computational methods [43,44,45], and the structure of several GTs with this fold, most of them belonging to family GT66, have been described [46]. These GTs usually contain an N-terminal transmembrane domain and a C-terminal globular domain with GT activity. In addition, the structure of the enzyme DUF1792 revealed a novel fold designated as GT-D type [47] and defined family GT101. Members of the GT51 family also have a distinct structure defined as lysozyme-type [48,49], and the recently published structure of TagA, a member of the GT26 family, suggests the definition of a novel fold termed GT-E [50]. A common feature found in many members of the GT-A and GT-C superfamilies is a DxD motif involved in binding to divalent metal ion and catalysis. However, the occurrence of this motif is not enough to predict a potential GT function since many GTs do not possess it and the DxD motif is found in more than a half of all described proteins [32]. Finally, given the modular nature of carbohydrate-active enzymes, a number of GTs contain noncatalytic domains potentially involved in substrate recognition or protein–protein interaction, and some polypeptides contain two GT modules belonging to the same or different families.

Table 1 shows a classification of GT families based on reaction mechanisms and fold types.

## 3. Glycosylation in Bacterial Pathogenesis

As mentioned in the introduction, effectors from pathogenic bacteria often exhibit specific enzymatic activity against host cell targets as part of its infection strategy. Protein glycosylation, the most common post-translational modification, has been traditionally thought to be restricted to eukaryotic organisms. However, in the last years, many examples of bacterial virulence factors and toxins exhibiting GT activity have been characterized [54]. Bacterial glycosylation is broadly linked to bacterial pathogenesis [55]. It fulfills two main functions during infection: firstly, protein glycosylation enables bacterial adhesion to host cells and, secondly, leads to modification of specific host targets with the consequent manipulation of host cellular processes by the pathogen to their own benefit.

Bacterial GTs perform two types of protein modifications: N-linked glycosylation or O-linked glycosylation [29]. Whereas N-GTs conjugate glycans to side chain amide nitrogen of an asparagine in an Asn-X-Ser/Thr consensus sequence, *O*-glycosylation happens on the hydroxyl group of a serine or threonine residue. Both N-linked and O-linked glycosylation pathways may modify multiple proteins. Interestingly, there are examples of pathogenic bacteria, like *C. rodentium*, performing *N*- and *O*-glycosylation that are both critically required for colonization of the gastrointestinal tract in infected mice [54].

In this section we provide representative examples of well characterized bacterial GTs, attending to their specific biological role during infection and the type of modification they perform.

### 3.1. Glycosylation of Adhesins

One essential step to initiate infection is bacterial adhesion to the host epithelium. This first step depends sometimes on glycosylation of bacterial adhesins. Some examples are described below.

A well-known example is the high-molecular-weight (HMW) system of *Haemophilus influenza*. HMW1/HMW2 are two adhesins that are glycosylated at the cytoplasm by GTs HMW1C and HMW2C, respectively [56,57]. Then, they are transported to the bacterial surface and mediate adherence to respiratory epithelial cells [57]. Glucose and galactose are added to specific asparagine residues in the eukaryotic-like Asn-X-Ser/Thr consensus [58].Pgl (from Protein glycosylation) system in *Campylobacter jejuni* is another example of a glycosylated adhesin that has been considered a prototype for the relevance of bacterial glycosylation in bacterial–host interaction [55]. It is an N-linked protein glycosylation system composed of several proteins, including various GTs that modify *C. jejuni* proteins leading to bacterial–host cell adhesion and invasion of host epithelium [59]. Disruption of the Pgl pathway results in reduction of human epithelial cells adhesion and decreases the ability to colonize mouse and chicken in vivo [59,60,61]. Modification catalyzed by the Pgl system consists of the addition of a heptasaccharide to asparagine residues within the eukaryotic-like Asn-X-Ser/Thr glycosylation consensus. The oligosaccharide is assembled on the cytoplasmic side of the inner membrane. Then, it is translocated by an ABC-transporter (PglK) [62]. Once in the periplasmic space, PglB, an oligo-saccharyltransferase, transfers the heptasaccharide to asparagine residues of target ligands [63,64].In addition to *N*-glycosylation-related mechanisms to improve bacterial adhesion, other systems based on *O*-glycosylation have been characterized. One example is the emerging bacterial autotransporter heptosyltransferase (BAHT) family found in several Gram-negative bacteria [65]. Autotransporters which belong to the T5SS family have a modular composition with an N-terminal passenger domain and a C-terminal β-barrel domain that drives the passenger across the outer membrane. The passenger domain may be heptosylated by a BAHT member on numerous serine residues. This modification occurs before translocation of the protein across the inner membrane and is required for bacterial adhesion to the host. Two well-characterized BAHT member are AAH (autotransporter adhesin heptosyltransferase), from diffusely adhering *E. coli* isolate 2787, and TibC from enterotoxigenic *E. coli* strain H10407 [65,66]. In addition to pathogenic *E. coli*, BAHT members have been identified in other bacteria, including *Citrobacter*, *Shigella*, *Salmonella,* and *Burkholderia* [65]. Further studies are necessary to fully characterize these interesting bacterial enzymes.Post-translational modification of bacterial proteins to enhance bacterial adhesion to the host is not restricted to Gram-negative bacteria. A well-known example of modified bacterial proteins in Gram-positive bacteria are the *O*-glycosylated serine-rich repeat proteins (SRRPs) [67]. These large glycoproteins, unique in Gram-positive bacteria, are crucial for biofilm formation and, more importantly, host cell adhesion [68]. The best-characterized SRRP is Fap1 from *Streptococcus parasanguinis*. Glycosylation of Fap1 relies on the activity of at least six different GTs, although the specific function and glycan transferred remains unclear [69]. This family of glycoproteins is highly conserved among streptococcal and staphylococcal species [67], thus, characterization of SRRPs GTs and understanding of their biological role are key to identify new therapeutic drug targets in order to tackle relevant Gram-positive pathogens.

### 3.2. Glycosylation of Flagellins

Glycosylation of flagellin is an important component of numerous flagellar systems in Archaea and Bacteria, playing an important role in assembly and virulence [70], since this organelle, in addition to conferring motility, can play a role in every step of the infection cycle [71]. Flagella *O*-glycosylation has been demonstrated in many polar flagellins from Gram-negative [72] and Gram-positive bacteria [73,74,75,76,77,78,79,80], and also in a limited number of lateral flagellins [72,81,82]. Bacteria can glycosylate flagellin at a varying number of acceptor sites, from a single site in *Burkholderia* spp. or *Listeria monocytogenes* to multiple sites in *C. jejuni* or *Selenomonas sputigena* [80,83,84,85]. Moreover, whereas some bacteria use a single monosaccharide, others show strain-to-strain glycan heterogenetity [72]. Selected examples are described below.

In *Aeromonas caviae* Sch3N, the polar flagellins, FlaA and FlaB, are glycosylated with pseudoaminic acid glycans and this glycosylation is required for flagellar assembly. Genes involved in glycosylation are mapped in the *O*-antigen biosynthetic cluster [86,87]. In *Aeromonas hydrophila* AH-3, lateral flagellin is modified with a derivative of pseudaminic acid, whereas polar flagella are glycosylated with a heterogeneous glycan [88].*C. jejuni* FlaA is glycosylated at up to 19 sites that contribute to the flagellum serospecificity. FlaB is also glycosylated, but the number of sites remains to be established. The predominant *O*-glycans are derivatives of pseudoaminic acid and legionaminic acid. The genes encoding GTs involved in flagellin glycosylation are located adjacent to the flagellin structural genes, *flaA* and *flaB* [85,89,90].In *Helicobacter pylori*, both the FlaA and FlaB structural proteins are glycosylated with pseudomainic acid at seven and ten sites, respectively. Biosynthesis of pseudaminic acid has been extensively studied in *H. pylori* and *C. jejuni* and involves six consecutive enzymatic steps before being transferred onto flagellins via O-linked glycosylation [91,92].The GT transferring pseudaminic acid onto flagellin has been studied in *Magnetospirillum magneticum* AMB-1 [93]. This GT, known as Maf, displays significant sequence identity with proteins from other Gram-negative bacteria, including Maf1 from *A. caviae* and PseE from *C. jejuni*, that have been shown to transfer pseudaminic acid onto flagellins in these organisms [94,95]. The structure of *M. magneticum* Maf displays a modified GT-A topology as found in CAZy families GT29 and GT42. In fact, the closest structural homologue is *C. jejuni* sialyltransferase Cst-II (PDB 1RO7), belonging to family GT42 [93].Glycosyltation of *Clostridium difficile* flagellin is essential for motility. Two types of modification have been defined as type A and B. Type A modification involves an O-linked N-acetylglucosamine (GlcNAc) linked to a methylated threonine via a phosphodiester bond. Three GTs are involved in type B modification: GT1 and GT2 are responsible for the sequential addition of a GlcNAc and two rhamnoses, respectively, and GT3 is associated with the incorporation of a sulfonated peptidyl-amido sugar moiety [74].

### 3.3. Modification of Host Cell Targets by Bacterial GTs

In addition to bacterial GTs that glycosylate their own proteins, we find another interesting group of bacterial proteins that modify host cell proteins, manipulating the cell physiology to counteract the host defense response. The large clostridial toxins TcdA/TcdB, the *Legionella* cytotoxic glycosyltransferase family (Lgt), and N-GTs from the NleB/SseK family are included in this group [54].

Large clostridial glycosylating toxins: pathogenesis of *Clostridium* depends on the activity of two similar toxins, TcdA and TcdB, that exhibit GT activity in their N-terminal domains [96,97]. Both toxins perform *O*-glycosylation of small GTPases of the Rho family, inhibiting the regulatory functions of these proteins [98]. This modification occurs on the side chain hydroxyl group of a threonine residue, which is specific for each targeted Rho GTPase [99]. Another example of this family of toxins recently identified is PaTox from entomopathogenic *Photorhabdus asymbiotica* [100].*Legionella pneumophila* Lgt family: *L. pneumophila* is the causative agent of legionellosis, a severe lung disease. This facultative intracellular pathogen uses a T4SS to inject more than 300 effectors into the host cell [101]. Among them, *L. pneumophila* secretes several GTs (Lgt1-3) that perform specific glycosylation of a serine residue (Ser_53_) in the eukaryotic elongation factor 1A, a large GTPase and a component of the elongation complex in protein synthesis. This modification results in inhibition of protein synthesis and subsequently causes the death of infected cells [102,103]. Interestingly, Lgt1 share high structural homology with the clostridial toxin TcdB. Both GTs are members of the clan III of retaining enzymes with the GT-A fold [103,104,105].NleB/SseK family: we refer to this family of T3SS effectors in the next section.

## 4. Glycosyltransferases of the NleB Family

### 4.1. Members of the Family

T3SS effectors that catalyze the transfer of GlcNAc to arginines of host proteins have been studied in *C. rodentium*, EHEC, EPEC, and *S. enterica* (Table 2).

NleB was identified in *C. rodentium* together with other potential type III effectors encoded outside the LEE. They were designated NleA, NleB, NleC, NleD, NleE, NleF, and NleG for non-LEE-encoded effectors [106]. Two homologs of NleB are encoded in the genomes of EHEC and EPEC: NleB1 and NleB2 [106,108,109]. NleB1 is 89% identical to NleB from *C. rodentium*, and there is a difference in seven amino acids between the EHEC and EPEC versions. NleB2 is identical in both *E. coli* pathotypes but is only 60% identical to *C. rodentium* NleB, although an ortholog of *nleB2* is present in the genome of *C. rodentium* as a pseudogene [107]. Interestingly, *nleB2* is a non-neutral evolving gene, probably due to positive selection, suggesting that this gene is still in the process of optimizing its function [114]. *S. enterica* strains encode three homologs of NleB: SseK1, SseK2, and SseK3 [111,112]. Although the genes encoding these effectors are not located in the pathogenicity islands where the cognate T3SS are encoded, all of them have evidence of horizontal transfer and some are associated with other pathogenicity islands or prophages (Table 2). Interestingly, whereas *sseK1* and *sseK2* are present in most available *Salmonella* genome sequences, the phage-encoded SseK3 effector has a limited distribution [111]. A BLAST search on complete genomes available at the National Center for Biotechnology Information (NCBI) reveals that proteins identical or very similar to NleB1 or NleB2 are encoded in different strains of the emerging A/E pathogen *Escherichia albertii* [115], and significantly similar proteins are found in *Escherichia marmotae* [116] and *Yersinia hibernica* [117].

Percentages of identity between representative NleB/SseK proteins are shown in Table 3, and a phylogenetic analysis is shown in Figure 4.

### 4.2. Expression, Translocation, and Subcellular Localization

Effectors of T3SSs are expected to be expressed in coordination with the synthesis of their cognate secretion apparatus. This is the case for effectors of the LEE, SPI1 or SPI2 systems that are encoded in the islands, but the issue is not so obvious for effectors encoded outside the islands, including the members of the NleB/SseK family.

In A/E organisms, Ler is the main transcriptional regulator of LEE. It is encoded in the island and its level is controlled by other transcriptional regulators that respond to different environmental stimuli, including host hormones, microbiota-liberated sugars and metabolites, molecular oxygen at the gut epithelium, and contact of bacteria with host epithelial cells [120]. Expression is undetectable when bacteria are grown in rich laboratory media, but rapid expression and secretion of proteins to the supernatant is observed upon transfer to MEM-HEPES medium, which is associated with the heterogeneous expression of the EspADB translocon apparatus [121]. Several *nle* genes, including *nleB*, are transcribed in EHEC under these secretion-permissive conditions, although induction of *nleB* expression was slight (twofold) and was not coordinated with the synthesis of EspA filaments at the single-cell level [122]. Secretion of NleB (NleB1 in EPEC) by the LEE-encoded T3SS has been demonstrated in *C. rodentium* and EPEC [106,110]. Transfection with GFP-fusions or translocation of TEM-1 β-lactamase fusions [123] from EPEC into HeLa cells reveals a diffuse cytosolic localization of NleB1 [110,124]. The regulatory region of EPEC *nleB2* contains a motif conserved among a subset of genes encoding effector proteins in A/E pathogens that was named NRIR (*nle*
regulatory inverted repeat). In contrast to other *nle* genes, like *nleA*, expression of *nleB2* is not regulated by Ler. However, it is repressed by another LEE-encoded regulator, GlrR, which is also a transcriptional repressor of the LEE, and by the global regulator H-NS [125]. An interesting observation is that similar to the LEE genes and *nleA*, *nleB2* is induced highly in DMEM (Dulbecco’s modified Eagle’s medium) cultures and repressed in LB (lysogeny broth) medium, suggesting that in spite of the different regulatory elements, expression of these genes is coregulated by environmental signals to ensure their coexpression with the T3SS [125]. Secretion of NleB2 was demonstrated by analyzing the secretome of EPEC using stable isotope labeling with amino acids in cell culture [126].

In *Salmonella*, many different environmental stimuli and regulatory proteins affect SPI1 expression. HilA is a direct transcriptional activator, and the *hilA* gene is transcriptionally activated by HilD, HilC, and RtsA, three AraC-like regulatory proteins that create a complex feed-forward regulatory loop [127] that function partially by relieving repression mediated by H-NS [128]. The SPI1-encoded T3SS is deactivated within host cell vacuoles after invasion of epithelial cells or uptake into macrophages [129]. Two systems involved in this repression are the Rcs phosphorelay system [130] and the PhoQ/PhoP two-component system [131,132], which is critical for adaptation of *Salmonella* to the intravacuolar environment. The PhoQ/PhoP system is activated in the presence of low levels of divalent cations and low pH, that are found inside the SCV [133,134]. Interestingly, this regulatory system positively controls the expression of T3SS2 through transcriptional and post-transcriptional activation of the SsrA/SsrB two-component system, the main positive regulator of SPI2 [135]. Optimal expression of T3SS1 and T3SS2 can be achieved in vitro using appropriate media and culture conditions: LB medium with high osmolarity and microaerophilic conditions for the T3SS1, and LPM minimal medium with low pH, low Mg^++^ concentration, and aeration conditions for the T3SS2. Significant amounts of SseK1 are produced under both sets of conditions [112], although transcription is higher under SPI2-inducing conditions [136]. This may be a consequence of the fact that transcription of the gene *sseK1* is not regulated by the main regulators of SPI1 or SPI2 (HilA, HilD, and SsrB), although it is directly activated by PhoP in a SsrB-independent manner. This pattern of expression is consistent with SseK1 being translocated through T3SS1 and T3SS2, with different patterns and kinetics depending on the specific host cell type (epithelial, macrophages, or fibroblasts) [136]. In contrast to SseK1, SseK2 and SseK3 were detected specifically under SPI2-inducing conditions and their expression was dependent on the SsrA/SsrB system, suggesting coordination with the expression of the T3SS2 [111,112]. T3SS2-dependent translocation of SseK2 into HeLa cells was shown using a CyaA fusion [112], whereas T3SS2-dependent secretion into the culture supernatant and translocation into HeLa cells of SseK3 was shown using an HA-tagged form of this effector [111]. Transfection of HeLa cells with a plasmid expressing a SseK1-GFP fusion indicated that it localized uniformly throughout the cytoplasm rather than associate with membrane components or other discrete structures within the host cell [112]. The same localization was found upon translocation of SseK1-HA into RAW264.7 macrophages [137]. In contrast, SseK2-HA and SseK3-HA were associated to the Golgi network when translocated into macrophages [137], a localization that was also observed after ectopic expression of GFP-SseK3 [138].

### 4.3. Enzymatic Activity and Structural Studies

Before the characterization of the activity of NleB/SseK effectors, *N*-glycosylation of arginine residues in proteins was rarely reported [139], and most studies on protein glycosylation had focused on the *O*-glycosylation of serine, threonine, and tyrosine residues or *N*-glycosylation of asparagine residues [55]. Therefore, the discovery that EPEC NleB1 has arginine GlcNAc transferase activity toward several proteins was surprising [140,141]. GlcNAcylation activity was previously reported for NleB from *C. rodentium*, although the specific target residues were not characterized [142]. More recently, arginine N-GlcNAcylation activity has been shown for *C. rodentium* NleB, EHEC NleB1, and *S. enterica* SseK1, whereas activity of NleB2 has not been detected and conflicting results exist about SseK2 and SseK3 [137,143].

The structures of several NleB/SseK family members were recently reported, indicating that these proteins possess the typical GT-A family fold and the metal-coordinating DxD motif essential for ligand binding and enzymatic activity [144,145,146,147]. However, some controversy exists about the mechanism of transfer used by these glycosyltransferases. A study whose objective was the synthesis of an antibody that can specifically recognize arginine *N*-GlcNAcylated peptides suggested a β-anomeric configuration for the products of the reaction [148], implying that these enzymes are inverting GTs. However, the crystal structure of SseK3, which was reported in its free form and bound to UDP, GlcNAc, and manganese, suggested that it was a retaining arginine GT [144]. This conclusion was confirmed by another study that investigated, by NMR spectroscopy, the glycosidic bond configuration of a GlcNAc-GAPDH_187-203_ glycopeptide formed enzymatically by incubation with SseK1 and UDP-GlcNAc/MnCl_2_ [145,149]. A more recent study determined the crystal structure of EPEC NleB1 alone, NleB1 in complex with the death domain of FAS-associated death domain protein (FADD, one of its substrates, see below), and the death domain of TRADD (TNFR1-associated death domain protein) and RIPK1 (receptor-interacting serine/threonine-protein kinase 1) GlcNAcylated by NleB1 [146]. This study concluded that NleB1 is an inverting enzyme that converts the α-configuration in the UDP-GlcNAc donor to the β configuration in the arginine-GlcNAc product. The same study also cast doubts on the validity of previous conclusions since the study on SseK3 did not involve the analysis of the modified substrate and the SseK1 study forced the modification of a GAPDH peptide that may not be a physiological substrate. Finally, another study reporting the structure of SseK1 and SseK3 indicated that the closest structural homologs are enzymes from families GT44 and GT88, both containing retaining GTs (Table 1), but the configurations of the modified substrates were not studied [147]. Structures of SseK3 and NleB1 are shown in Figure 5.

### 4.4. Role in Virulence

The role in virulence of particular T3SS effectors is often difficult to demonstrate due partially to the fact that many other effectors and virulence factors are involved, sometimes with opposite or potentially redundant effects. Apparent redundancy can also be a consequence of limitations of laboratory models of pathogenesis [150]. In the case of NleB/SseK effectors, involvement in virulence has been shown for some members of the family.

A signature-tagged mutagenesis screen in C57BL/6 mice identified NleB as an important virulence factor of *C. rodentium*: deletion of *nleB* resulted in reduced colonization of mice, with a competitive index (CI) in the colon of 0.069, and reduced colonic hyperplasia [110]. The GT activity of NleB is very relevant for this phenotype, since a mutant with an altered DxD catalytic site, NleB_221_DAD-AAA_223_, failed to complement the attenuation of the *∆nleB* mutant. However, the fact that the catalytic mutant was able to provide partial complementation suggests that NleB may have functions that are independent of its GT activity [151]. Additional complementation experiments showed that EHEC NleB1 was able to restore the virulence of the *C. rodentium ∆nleB* mutant, whereas NleB2 was unable to complement the colonization defect. This study also identified a structural motif in NleB1, His_242_-Glu_253_-Asn_254_, which was essential for activity and virulence [145]. The Glu_253_ residue, together with Tyr_219_ were identified in a previous study as essential for enzymatic activity and virulence function of NleB1 [152]. Interestingly, the presence of *nleB1* in EHEC and EPEC strains is associated with colonization of sheep [153] and cattle [154], with human disease [155,156,157,158,159], and with an increase in disease severity [160,161,162,163,164,165]. In contrast, the role of NleB2 in virulence is currently unclear.

The first study describing *S. enterica* serovar Typhimurium effectors SseK1 and SseK2 failed to detect signs of attenuation for *∆sseK1*, *∆sseK2*, or *∆sseK1 ∆sseK2* mutants during infection of tissue culture cells or susceptible (Nramp1^s^) BALB/c mice using a time-to-death assay after intraperitoneal infections [112]. Another study showed that the *∆sseK1 ∆sseK2 ∆sseK3* triple mutant had significantly reduced growth levels in RAW264.7 macrophages but confirmed the lack of attenuation in mice [166]. A more sensitive assay using mixed infections of mice identified a mild but statistically significant attenuation for the *sseK1 sseK2* double mutant after oral infections (CI = 0.4–0.5 compared with the wild type). No difference was detected between the *sseK1 sseK2 sseK3* triple mutant and the *sseK1 sseK2* double mutant, suggesting that SseK3 has no role in virulence in this system [111]. Mild attenuation was also reported for an *sseK1* single mutant of *S. enterica* serovar Typhimurium after intraperitoneal and oral infections of BALB/c mice using the CI analysis [136]. In addition, an *sseK1* deletion mutant of *S. enterica* serovar Enteritidis showed decreased formation of biofilm, reduced intracellular survival in activated mouse peritoneal macrophages, and reduced pathogenicity in BALB/c mice [167]. The same group has recently shown that a *∆sseK2* mutant of *S. enterica* serovar Typhimurium is also defective in biofilm formation and virulence [168]. Interestingly, a genomic screen for genes involved in long-term systemic infection in 129X1/SvJ (Nramp1^r^) mice detected *sseK2* as one of the genes contributing to serovar Typhimurium infection by 28 days postinfection. The CI of the *sseK2* mutant in this system was 0.0041 in mesenteric lymph nodes, and 0.0026 in spleen [169]. This study underlines the importance of finding the right model to highlight the role of a given effector in virulence.

### 4.5. Targets of NleB/SseK Effectors

The most clearly demonstrated function of this family of T3SS effectors is the inhibition of NF-κB pathway activation and of host cell death during infection. A first report suggested that NleB1 from EPEC contributed to NF-κB inhibition by enhancing the activity of NleE, another effector that inhibits IκB phosphorylation, leading to the stabilization of IκB that retains NF-κB in the cytoplasm [170]. Another study proposed a model where NleB1 acted at a different, upstream point in the NF-κB signaling pathway that is activated in response to TNFα [124]. Involvement in the TNF receptor (TNFR) pathway, but not the TLR/ILR1 pathway, was confirmed using specific luciferase reporters [171]. This is a death receptor pathway that involves homo- or heterotypic interactions between intracellular death domains in receptors and death domains in the downstream adaptor proteins, including TRADD and FADD [172]. The mechanism of action of NleB1 was elucidated by two outstanding reports that used a yeast two-hybrid screen of a HeLa cDNA library and co-immunoprecipitation assays to identify host binding partners. NleB1 directly inactivated death domains in several proteins including TRADD, FADD, RIPK1, and TNFR1 [140,141]. This inhibition depended on the N-GlcNAc transferase activity of NleB1, which specifically modified Arg_235_ and Arg_117_ in the death domains of TRADD and FADD, respectively. These modifications blocked interactions between death domains, thereby disrupting inflammatory NF-κB signaling, caspase 8-dependent apoptosis, and necroptosis [173]. NleB from *C. rodentium* showed the same activity, and the mouse model of infection with this A/E pathogen demonstrated that the GlcNAc transferase activity was crucial for virulence [140,141]. NleB2 from EPEC showed a lower enzymatic activity and only a fraction of TRADD was GlcNAcylated after transfection of HEK293T cells with a plasmid expressing NleB2 [140]. A more recent study combining an arginine-GlcNAc-specific enrichment method coupled with mass spectrometry concluded that the preferred target of EPEC NleB1 and *C. rodentium* NleB was FADD, and that overexpression of these effectors led to GlcNAcylation of additional proteins that may be nonphysiological targets [174].

At least some of the members of the family present in *S. enterica* serovar Typhimurium have an effect on the NF-κB pathway. Similar to NleB1, SseK1 inhibited TNF-α-NF-κB signaling and catalyzed GlcNAcylation of TRADD after transfection of host cells [140]. Also, ectopic expression of SseK3 blocked TNF-stimulated NF-κB activity and this effect depended on the conserved DxD catalytic motif [138]. Importantly, SseK1 and SseK3 inhibited *Salmonella*-induced NF-κB activation and necroptotic cell death during infection in macrophages [137]. Although SseK2 was also able to inhibit NF-κB activation after TNF-α stimulation, it is not clear if it is a bona fide inhibitor of this pathway since the effect was detected only after ectopic overexpression in mammalian cells but not during macrophage infections. Expression of SseK1 or SseK3 in a *∆sseK1/2/3 Salmonella* mutant induced different arginine-GlcNAcylation patterns during infection of RAW264.7 macrophages, whereas expression of SseK2 did not result in detectable modifications. Furthermore, the death domain protein FADD was GlcNAcylated by SseK1, but not by SseK2 or SseK3, and the death domain protein TRADD was modified by SseK1 and, to a lesser extent, by SseK3 [137]. Another study confirmed that SseK1, SseK3, EHEC NleB1, EPEC NleB1, and *C. rodentium* NleB, but not SseK2 or NleB2, blocked TNF-mediated NF-κB pathway activation. However, in conflict with the previous report, the same study reported glycosylation of FADD by SseK2, but not by SseK1 or SseK3 [143]. A recent study combined arginine-GlcNAc glycopeptide immunoprecipitation and mass spectrometry to identify host proteins GlcNAcylated by endogenous levels of SseK1 and SseK3 during *Salmonella* infection. This study confirmed modification of TRADD, but not FADD, by SseK1 and described the death domains of receptors TNFR1 and TRAILR as physiological SseK3 targets that were modified at residues Arg_376_ and Arg_293_, respectively [147].

Some proteins that do not contain death domains have also been described as targets of NleB/SseK effectors. An early paper identified glyceraldehyde 3-phosphate dehydrogenase (GAPDH) as a binding partner and substrate of the glycosyltransferase activity of *C. rodentium* NleB [151]. The same study described the interaction of GAPDH with the TNFR-associated factor 2 (TRAF2), a protein required for TNF-α-mediated NF-κB activation. Data presented in this report suggested a model in which GlcNAcylation of GAPDH by NleB would disrupt the TRAF2-GAPDH interaction to suppress TRAF2 polyubiquitination and NF-κB activation. Another study from the same authors suggested that this modification of GAPDH also inhibited the interaction with TRAF3 and reduced Lys_63_-linked TRAF3 ubiquitination, leading to inhibition of type I IFN signaling [142]. Although several researchers were unable to detect NleB-dependent GlcNAcylation of GAPDH [140,146,174], this modification was confirmed by other studies [143,145,175], and mass spectrometry and site-directed mutagenesis demonstrated that EHEC NleB1 glycosylates GAPDH on Arg_197_ and Arg_200_ [143]. In addition, GAPDH was also shown to be a target for SseK1 [143].

A new role has been recently suggested for EPEC NleB1 through its interaction and modification of HIF-1α, a master regulator of cellular O_2_ homeostasis. NleB1 catalyzes GlcNAcylation at Arg_18_ at the N-terminus of HIF-1α, thereby enhancing its transcriptional activity and inducing downstream gene expression to alter host glucose metabolism [176].

There are also host binding partners described for NleB/SseK proteins whose GlcNAcylation has not been demonstrated. This is the case of TRIM32, the first binding partner identified for SseK3 [138]. TRIM32 is an E3 ubiquitin ligase that is not required for SseK3 to inhibit NF-κB signaling during infection of macrophages [137]. DRG2 (developmentally-regulated GTP-binding protein 2), LRRC18 (leucine-rich repeat-containing protein 18), and POLR2E (DNA-directed RNA polymerases I, II, and III subunit RPABC1), identified in a yeast two-hybrid screen [177], and MAP7 (ensconsin), identified by a quantitative proteomics approach and confirmed by coimmunoprecipitation [178], are novel binding partners of EHEC NleB1, although the GlcNAcylation of these proteins and the physiological role of these interactions have not been shown. A summary of host targets of NleB/SseK effectors is depicted in Figure 6.

Interestingly, a recent report has shown that NleB1 can also glycosylate a bacterial protein, GshB (glutathione synthetase) on Arg_256_. In fact, this was the most abundant arginine-GlcNAcylated target enriched from samples of HEK293T cells infected with EHEC [179]. GshB was also glycosylated in vitro and in vivo by *C. rodentium* NleB and in vitro by *Salmonella* SseK1. In vitro glycosylation of GshB enhances GshB-mediated production of glutathione, which provides resistance to oxidative stress. Consistent with that, both the *nleB* and *gshB* mutants of *C. rodentium* had a significant growth defect in the presence of H_2_O_2_, suggesting a new role for NleB in promoting survival of these bacteria in peroxide stress conditions.

## 5. Conclusions

T3SS effectors are included in the arsenal of weapons that some Gram-negative pathogenic bacteria use to counteract the host immune system and succeed in infection. After millions of years of coevolution, many pathogenic bacteria have acquired the capacity to manipulate host cellular processes by mimicking the biological function of eukaryotic proteins. Identification of interacting partners and deciphering of the molecular mechanisms that these fascinating molecules use to subvert host defenses are essential to design new antibacterial strategies. Exploiting host functions by those T3SS effectors is a very exciting example of the co-evolutionary battle of host–pathogens interactions. The NleB/SseK family of effectors constitute a good illustration of this. The discovery of a family of bacterial T3SS effectors able to catalyze the N-linked GlcNAcylation of arginine residues was surprising since protein glycosylation usually occurs in a serine or threonine residue (O-linked glycosylation) or in an asparagine residue (N-linked glycosylation). This small modification has a great impact in the interaction of A/E pathogens and *Salmonella* since it affects a critical residue for death domain interactions. The subsequent inhibition of NF-κB activation and cell death could provide a way to counteract host immune responses and to prolong attachment of A/E bacteria to the gut epithelium. Besides this well-established role of the NleB/SseK family of effectors, other targets have been described that need further research to understand the physiological consequences of the glycosylation of these host proteins during infections. Also the activity of NleB2 and SseK2 requires further evaluation since some studies suggest that these effectors may lack glycosylation activity or may have evolved to preferentially catalyze GlcNAcylation of a different amino acid residue or to transfer a different sugar [137,143]. There are also interacting partners that apparently are not glycosylated. What is the biological significance of these interactions? What are the consequences of them in the host or in the activity and/or localization of NleB/SseK proteins? Especially intriguing is the finding of a bacterial target for the enzymatic activity of these effectors because it opens the possibility of a dual role in interfering host processes and modulating the activity of endogenous proteins. The catalog of interacting partners and catalytic substrates will probably increase in the near future giving us a more complete picture of the functions of these bacterial effectors that may be exploited as possible targets of new therapeutic agents.

## Figures and Tables

**Figure 1 microorganisms-08-00357-f001:**
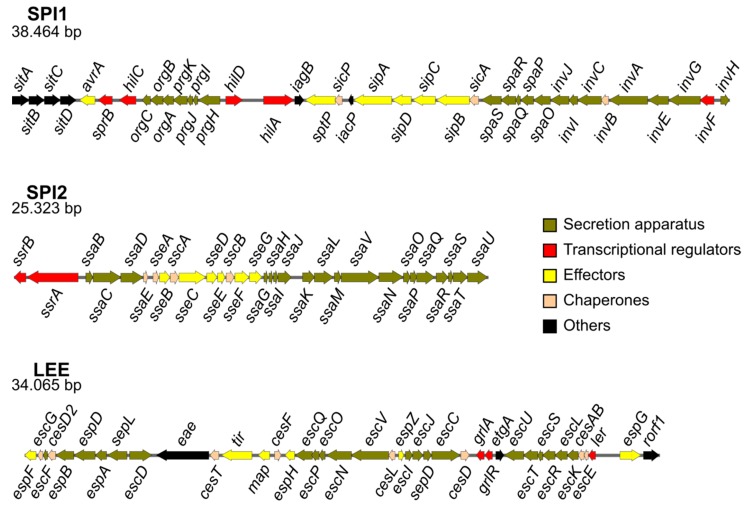
Genetic structure of *Salmonella* pathogenicity island 1 (SPI1) and *Salmonella* pathogenicity island 2 (SPI2) of *S. enterica* serovar Typhimurium strain 14028, and the locus of enterocyte effacement (LEE) island of *E. coli* O127:H6 strain E2348/69. Genes are colored according to their functional categories [22,28].

**Figure 2 microorganisms-08-00357-f002:**
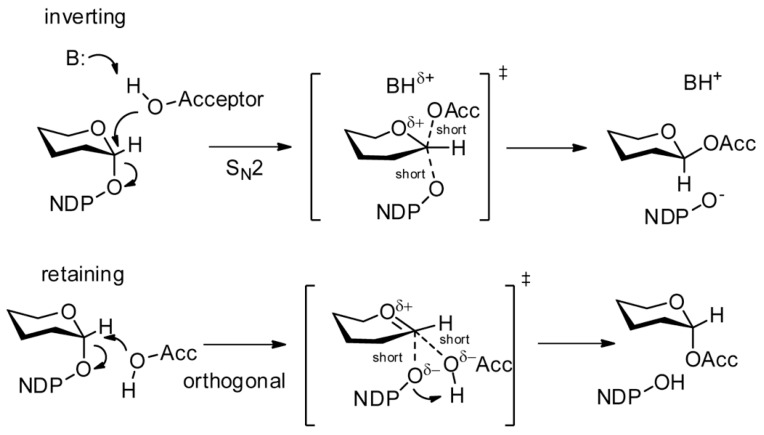
Glycosyltransferase mechanisms. An S_N2_ (substitution, nucleophilic, bimolecular) process is accepted for inverting glycosyltransferases (GTs). Several mechanisms have been proposed for retaining enzymes. The orthogonal mechanism is depicted here. ‡: Transition state. Image by Dr. Brock Schumann (Wikimedia Commons, CC BY-SA 3.0, https://creativecommons.org/licenses/by-sa/3.0/deed.en).

**Figure 3 microorganisms-08-00357-f003:**
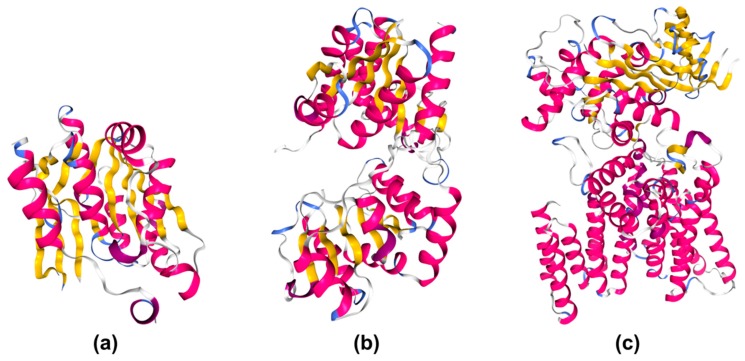
Glycosyltransferase folds. (**a**) Diagram of the GT-A fold protein SpsA from *Bacillus subtilis,* belonging to family GT2 (PDB ID: 1QG8) [51]; (**b**) diagram of the GT-B fold-type protein GtfB from *Amycolaptosis orientalis*, belonging to family GT1 (PDB ID: 1IIR) [52]; (**c**) diagram of the GT-C fold protein PglB from *Campylobacter lari*, belonging to family GT66 (PDB ID: 3RCE) [46]. Drawings created with NGL [53].

**Figure 4 microorganisms-08-00357-f004:**
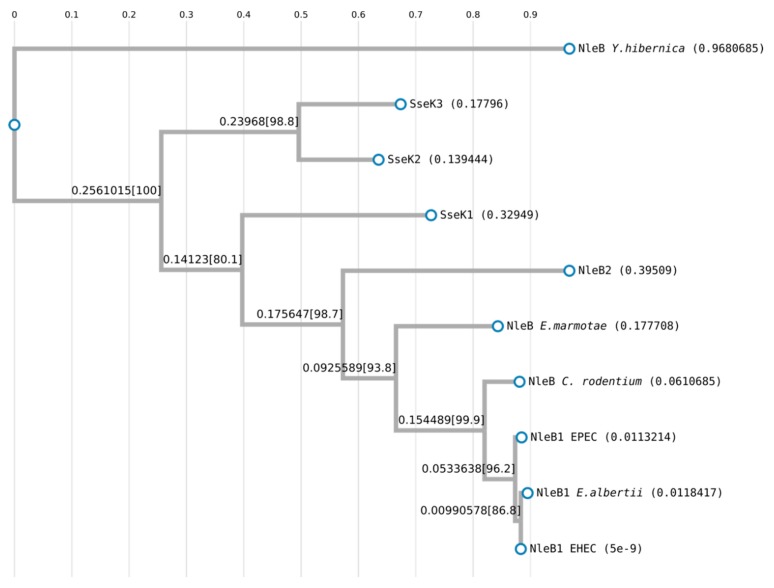
Phylogenetic analysis of NleB/SseK proteins. Alignment and phylogenetic reconstructions were performed using the function “build” of ETE3 v3.1.1 [118] as implemented on the GenomeNet (https://www.genome.jp/tools/ete/). The tree was constructed using FastTree v2.1.8 with default parameters [119]. Values at nodes are SH-like local support.

**Figure 5 microorganisms-08-00357-f005:**
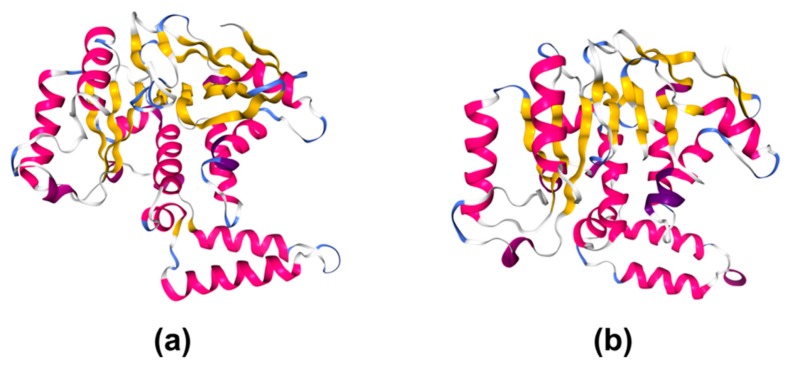
Structure of NleB/SseK family members. (**a**) Crystal structure of SseK3 from *S. enterica* serovar Typhimurium strain SL1344 (PDB ID: 6CGI) [147]; (**b**) crystal structure of NleB1 from *E. coli* O127:H6 (PDB ID: 6E66) [146]. Drawings created with NGL [53].

**Figure 6 microorganisms-08-00357-f006:**
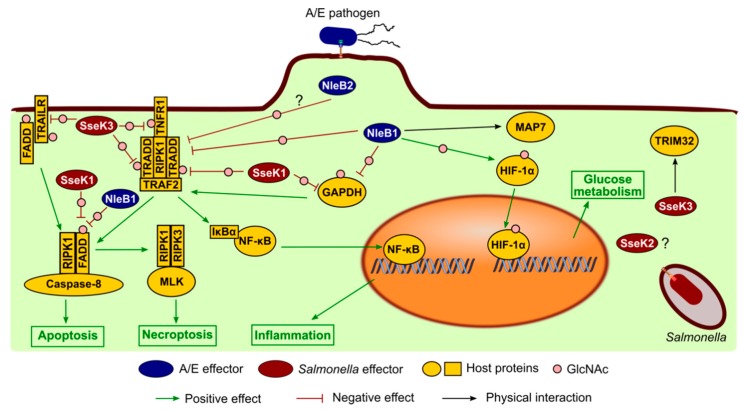
Summary of NleB/SseK host protein targets.

**Table 1 microorganisms-08-00357-t001:** Classification of glycosyltransferase families.

		Reaction Mechanism		
		Inverting	Retaining	Unknown
**Fold type**	*GT-A*	2 ^1^, 7, 12, 13, 14, 16, 21, 29, 31, 42, 43, 82, 84, 109	6, 8, 15, 24, 27, 34, 44, 55, 62, 64, 78, 81, 88	
	*GT-B*	1, 9, 10, 18, 19, 23, 28, 30, 33, 38, 41, 52, 63, 65, 68, 70, 80, 90, 104	3, 4, 5, 20, 35, 52, 72, 99, 107	
	*GT-C*	22, 39, 50, 53, 57, 58, 59, 66, 83, 85, 87		
	*Others*	26 (GT-E), 51 (lysozyme-type),		101 (GT-D)
	*Unknown*	11, 17, 25, 37, 40, 47, 48, 49, 54, 56, 61, 67, 73, 74, 75, 76, 92, 94, 97, 98, 100, 102, 103, 105, 106, 108	32, 45, 60, 69, 71, 77, 79, 89, 93, 95, 96	91, 110

^1^ Numbers correspond to the GT family numbers assigned in the carbohydrate active enzymes (CAZy) database.

**Table 2 microorganisms-08-00357-t002:** NleB/SseK glycosyltransferases.

Organism.	Reference Strain	Effector Name	Protein Size (aa)	Genomic Context	References
*C. rodentium*	ICC168	NleB	329	Genomic island GI4	[106,107]
Enterohemorrhagic *E. coli* (EHEC) O157:H7	EDL933	NleB1	329	O-island 122/SpLE3	[106,108]
		NleB2	326	O-island 36/Phage CP-933K/Sp3	
Enteropathogenic *E. coli* (EPEC) O127:H6	E2348/69	NleB1	329	Integrative element IE6	[109,110]
		NleB2	326	Phage PP4	
*S. enterica* serovar Typhimurium	14028	SseK1	336		[111,112,113]
		SseK2	348		
		SseK3	335	Phage ST64B	

**Table 3 microorganisms-08-00357-t003:** Percent identity data for NleB/SseK proteins provided by ClustalW.

	NleB *C. rodentium*	NleB1 EPEC	NleB1 EHEC	NleB *E. albertii*	NleB *E. marmotae*	NleB2	SseK1	SseK2	SseK3
NleB *C. rodentium*									
NleB1 EPEC	88.75								
NleB1 EHEC	89.06	97.87							
NleB1 *E. albertii*	88.45	97.57	99.09						
NleB *E. marmotae*	71.42	71.73	71.73	71.43					
NleB2 EPEC/EHEC	59.82	61.04	61.35	60.43	60.43				
SseK1 *S. enterica*	57.45	58.05	59.57	58.97	58.97	53.68			
SseK2 *S. enterica*	52.58	52.28	53.19	52.58	51.06	46.93	55.36		
SseK3 *S. enterica*	51.98	53.19	53.50	52.89	54.10	47.55	54.93	75.22	
NleB *Y. hibernica*	32.52	32.83	34.04	34.04	34.04	33.13	35.12	33.14	33.43

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
