# Peer review of "Type III Secretion Effectors with Arginine N-Glycosyltransferase Activity"

_microorganisms, 2020, doi:10.3390/microorganisms8030357_

Round 1

Reviewer 1 Report

Well written, comprehensive, detailed and insightful T3SS review with focus on glcosyltransferases. 

Small typos such as:

Line 39:  Plural of Salmonella is Salmonellae in Latin.

Lines 168, 170 and 177:  adhesins instead of adhesines

Line 311:  their expression instead of its expression

There are likely more I missed.

I would not mind if the authors would expand their discussion in the Conclusions section and provide their own creative perspective about the biological significance of the molecular/biochemical mechanisms described in a "bigger, more comprehensive picture or vision".

Author Response

Responses to reviewer 1:

Well written, comprehensive, detailed and insightful T3SS review with focus on glcosyltransferases.

Response: We thank the reviewer for the corrections suggested.

Small typos such as:

Line 39:  Plural of Salmonella is Salmonellae in Latin.

Response: -Typo corrected as suggested in line 39.

Lines 168, 170 and 177:  adhesins instead of adhesines

Response: -Typos corrected as suggested in lines 168, 170 and 177.

Line 311:  their expression instead of its expression

Response: -“Their” instead of “its” as suggested in line 349.

There are likely more I missed.

Response: -The manuscript has been checked and some additional typos or mistakes have been corrected:

-Line 83 and line 111: numbering of subsection: 2.1 instead of 3.1 and 2.2 instead of 3.2.

-Line 174: “asparagine residues” instead of “asparagines residues”.

-Line 179: “composed of” instead of “composed on”.

-Line 203: “O-Glycosylated” changed to “O-glycosylated”.

I would not mind if the authors would expand their discussion in the Conclusions section and provide their own creative perspective about the biological significance of the molecular/biochemical mechanisms described in a "bigger, more comprehensive picture or vision".

Response: -The conclusions section has been slightly expanded in lines 519-526 as follows:

“T3SS effectors are included in the arsenal of weapons that some Gram-negative pathogenic bacteria use to counteract the host immune system and succeed in infection. After millions of years of coevolution, many pathogenic bacteria have acquired the capacity to manipulate host cellular processes by mimicking the biological function of eukaryotic proteins. Identification of interacting partners and deciphering of the molecular mechanisms that these fascinating molecules use to subvert host defenses are essential to design new antibacterial strategies. Exploiting host functions by those T3SS effectors is a very exciting example of the co-evolutionary battle of host-pathogens interactions. The NleB/SseK family of effectors constitute a well illustration of this.”

Reviewer 2 Report

Major suggestions:

There is an extensive amount of literature on O-glycosylation of bacterial flagellins, which the authors do not mention. It would be useful to include some information on these GTs.

Minor suggestions:

Lines 191, 337, 371 – change ‘that’ to ‘which’

Line 193 – add ‘The’ before ‘passenger’

Line 194 – Does the modification occur before translocation across the inner membrane (i.e., in cytoplasm) or the outer membrane (i.e., in periplasmic space)?

Line 195 – change to “A well characterized BAHT member is AAH (autotransporter adhesion heptosyltransferase) from ….”

Line 198 – change ‘in other bacteria like’ to ‘in other bacteria, including’

Line 205 – remove comma after Fap1

Line 206 – remove commas on either side of ‘at least’

Lines 274 and 318 – change ‘associated to’ to ‘associated with’

Line 285 – change sentence to “An interesting observation is that similar to the LEE genes and nleA, nleB2 is induced highly in DMEM cultures and repressed in LB medium, suggesting that in spite of the different regulatory elements, expression of these genes is coregulated …”  I’m not sure what is meant by ‘LB rich medium’. It would also be helpful to define ‘DMEM cultures’.

Line 302 – again, what is meant by ‘LB rich medium’?

Author Response

Responses to reviewer 2:

Major suggestions:

There is an extensive amount of literature on O-glycosylation of bacterial flagellins, which the authors do not mention. It would be useful to include some information on these GTs.

Response: -We thank the reviewer for the suggestion. Some information about glycosylation of bacterial flagellins is now included in a new subsection: “3.2 Glycosylation of flagellins” in lines 211-247 that cite 26 articles that have been added to the list or references.

Minor suggestions:

Lines 191, 337, 371 – change ‘that’ to ‘which’

Response: “that” has been changed to “which” in lines 191, 375 and 409 of the revised version of the manuscript.

Line 193 – add ‘The’ before ‘passenger’

Response: “The” added in line 193.

Line 194 – Does the modification occur before translocation across the inner membrane (i.e., in cytoplasm) or the outer membrane (i.e., in periplasmic space)?

Response: It occurs before translocation across the inner membrane. This information has been added in lines 194-195.

Line 195 – change to “A well characterized BAHT member is AAH (autotransporter adhesion heptosyltransferase) from ….”

Response: Since two examples of BAHT members are mentioned the sentence has been changed into: “Two well characterized BAHT member are AAH (autotransporter adhesin heptosyltransferase), from diffusely adhering E. coli isolate 2787, and TibC from…” in lines 195-197.

Line 198 – change ‘in other bacteria like’ to ‘in other bacteria, including’

Response: Changed as suggested in line 198.

Line 205 – remove comma after Fap1

Response: Comma removed after Fap1 in line 205.

Line 206 – remove commas on either side of ‘at least’

Response: Commas removed in line 206.

Lines 274 and 318 – change ‘associated to’ to ‘associated with’

Response: “associated to” changed to “associated with” in lines 286 and 311.

Line 285 – change sentence to “An interesting observation is that similar to the LEE genes and nleA, nleB2 is induced highly in DMEM cultures and repressed in LB medium, suggesting that in spite of the different regulatory elements, expression of these genes is coregulated …”  I’m not sure what is meant by ‘LB rich medium’. It would also be helpful to define ‘DMEM cultures’.

Response: DMEM and LB are now defined in this sentence. The sentence in lines 322-325 has been changed into: “An interesting observation is that similar to the LEE genes and nleA, nleB2 is induced highly in DMEM (Dulbecco’s modified Eagle’s medium) cultures and repressed in LB (lysogeny broth) medium, suggesting that in spite of the different regulatory elements, expression of these genes is coregulated…”.

Line 302 – again, what is meant by ‘LB rich medium’?

Response: “LB rich medium” has been changed to “LB medium” in line 340.

Reviewer 3 Report

This is an excellent, comprehensive review of T3SS effectors with arginine N-glycosyltransferase activity.

The authors have done a very nice job with citing all of the relevant literature and have provided good balance in their description of conflicting findings.

I found only a few minor changes that should be made, although I would strongly advise the authors to proofread the grammar one more time:

Line 89: change '620000' to '620,000'

Line 90: change '13000' to '13,000'

Lines 176-177: change 'another example of glycosylated adhesines' to 'another example of a glycosylated adhesin'

Line 191: change 'Autotransporters, that belongs to the T5SS family, have' to 'Autotransporters that belong to the T5SS family have'

Line 193: change 'Passenger domain' to 'The passenger domain'

Line 226: change 'to the host cell' to 'into the host cell'

Line 236: change 'has been studied' to 'have been studied'

Line 243: change 'orthologous' to 'ortholog'

Line 472: reference 153 is incorrect. It should reference PMID 31974499

Author Response

Responses to reviewer 3:

This is an excellent, comprehensive review of T3SS effectors with arginine N-glycosyltransferase activity.

The authors have done a very nice job with citing all of the relevant literature and have provided good balance in their description of conflicting findings.

I found only a few minor changes that should be made, although I would strongly advise the authors to proofread the grammar one more time:

Response: We thank the reviewer for the corrections. Some additional changes have been introduced:

-Line 83 and line 111: numbering of subsection: 2.1 instead of 3.1 and 2.2 instead of 3.2.

-Line 174: “asparagine residues” instead of “asparagines residues”.

-Line 179: “composed of” instead of “composed on”.

-Line 203: “O-Glycosylated” changed to “O-glycosylated”.

 Line 89: change '620000' to '620,000'

Response: “620000” changed to “620,000” in line 89.

Line 90: change '13000' to '13,000'

Response: “13000” changed to “13,000” in line 90.

Lines 176-177: change 'another example of glycosylated adhesines' to 'another example of a glycosylated adhesin'

Response: “another example of glycosylated adhesines” changed to “another example of a glycosylated adhesin” in lines 176-177.

Line 191: change 'Autotransporters, that belongs to the T5SS family, have' to 'Autotransporters that belong to the T5SS family have'

Response: “Autotransporter, that belongs to the T5SS family, have” changed to “Autotransporters which belong to the T5SS family have” in line 191 of the revised manuscript.

Line 193: change 'Passenger domain' to 'The passenger domain'

Response: “Passenger domain” changed to “The passenger domain” in line 193.

Line 226: change 'to the host cell' to 'into the host cell'

Response: “to the host cell” changed to “into the host cell” in line 263.

Line 236: change 'has been studied' to 'have been studied'

Response: “has been studied” changed to “have been studied” in line 273.

Line 243: change 'orthologous' to 'ortholog'

Response: “orthologous” changed to “ortholog” in line 280.

Line 472: reference 153 is incorrect. It should reference PMID 31974499

Response: We thank the reviewer for signaling this error. The correct reference is now included as reference number 179 in line 510.